# Sustainable One-Pot Immobilization of Enzymes in/on Metal-Organic Framework Materials

**M. Asunción Molina [1], Victoria Gascón-Pérez [2], Manuel Sánchez-Sánchez [1],* and Rosa M. Blanco [1],***

1 Instituto de Catálisis y Petroleoquímica (ICP), CSIC, 28049 Madrid, Spain; asuncion.molina@csic.es
2 Department of Chemical Sciences, Bernal Institute, University of Limerick, V94 T9PX Limerick, Ireland; Victoria.GasconPerez@ul.ie
* Correspondence: manuel.sanchez@icp.csic.es (M.S.-S.); rmblanco@icp.csic.es (R.M.B.); Tel.: +34-915854795 (M.S.-S.); +34-915854636 (R.M.B.)

**Abstract:** The industrial use of enzymes generally necessitates their immobilization onto solid supports. The well-known high affinity of enzymes for metal-organic framework (MOF) materials, together with the great versatility of MOFs in terms of structure, composition, functionalization and synthetic approaches, has led the scientific community to develop very different strategies for the immobilization of enzymes in/on MOFs. This review focuses on one of these strategies, namely, the one-pot enzyme immobilization within sustainable MOFs, which is particularly enticing as the resultant biocomposite Enzyme@MOFs have the potential to be: (i) prepared in situ, that is, in just one step; (ii) may be synthesized under sustainable conditions: with water as the sole solvent at room temperature with moderate pHs, etc.; (iii) are able to retain high enzyme loading; (iv) have negligible protein leaching; and (v) give enzymatic activities approaching that given by the corresponding free enzymes. Moreover, this methodology seems to be near-universal, as success has been achieved with different MOFs, with different enzymes and for different applications. So far, the metal ions forming the MOF materials have been chosen according to their low price, low toxicity and, of course, their possibility for generating MOFs at room temperature in water, in order to close the cycle of economic, environmental and energy sustainability in the synthesis, application and disposal life cycle.

**Keywords:** Enz@MOF; enzyme immobilization; sustainable MOFs as supports; in situ; one-step; room temperature; low leaching; ZIF-8; Fe-BTC; NH$_2$-MIL-53(Al)



## 1. Scope of this Review

In the last decade, a huge number of metal-organic framework (MOF) materials, enzymes and strategies have been reported as suitable for generating enzyme@MOF biocomposites. This review covers just one of these approaches, specifically that where the solid biocatalysts are formed by the synthesis of a MOF material, acting as a support, in the presence of the enzyme to be immobilized. In other words, this review addresses the strategies for forming enzyme@MOFs known as 'one-pot', one-step, in situ or de novo methods. In recognition of the diversity of this field, these terms will be used interchangeably throughout this review. Readers interested in a complete literature review on enzymes are encouraged to check out recent reviews with wider scopes [1,2].

## 2. The Origins and Rising Dominance of the Enzyme-Supporting MOF

Enzyme immobilization is a topic with more than half a century of history [3], with enzymes showing superb advantages that were previously unattainable in industry due to issues with solubility and lability. Apart from methods to achieve enzyme insolubilization, the immobilization on solid supports has been the most widely studied strategy. Much effort has been made since then and thousands of materials have been studied as supports for the immobilization of enzymes, either covalently [4] or non-covalently [5,6]. Porous supports offer an extra incentive for enzyme immobilization, as they may ideally trap

enzyme molecules without modifying their structure or their active centers. Table 1 compares some of the most relevant physicochemical properties and performance of some selected porous enzyme supports with different strategies.

**Table 1.** Comparison of properties and performance of some selected immobilization strategies and porous supports of enzymes: covalent immobilization on amorphous agarose; non-covalent immobilization by post-synthetic or in situ addition to siliceous mesoporous ordered materials (MMO), and in situ immobilization onto MOFs. Table entries are displayed according to the following color-coding: green for good, yellow for middle and red for bad.

| - | Covalent Post-Synthetic (Agarose) [4,7] | Non-Covalent Post-Synthetic (MMO) [8] | Non-Covalent In Situ (MMO) [9,10] | Non-Covalent In Situ (MOFs) [11,12] |
|---|---|---|---|---|
| Surface area | Low: $\approx$200 $m^2/g$ | Moderate/high: $\approx$700 $m^2/g$ | Moderate/high $\approx$700 $m^2/g$ | Very high: >1000 $m^2/g$ |
| Pore width | >20 nm | $\approx$7–10 nm | $\approx$4–12 nm | Micropores < 2 nm |
| Pore connectivity | Amorphous: low | Excellent | Excellent | Excellent |
| Chemical affinity | Essential | Necessary | Unnecessary | Beneficial |
| Activity preserved | Low/moderate | High/moderate | High/moderate | High/moderate |
| Enzyme loading | Moderate/high | Moderate/high | Moderate/high | Moderate/high |
| Enzyme leaching | None | Low | Very low | Negligible |
| Enzyme stabilization | High | Moderate/high | Moderate | Moderate/high |

Simply anchoring an enzyme to a support is relatively easy and, in many cases, just enough to catalyze a reaction successfully, but optimizing the biocatalyst and understanding what happens to the immobilized enzyme may be difficult. Covalent immobilization involves the chemical modification of the enzyme, which often leads to decreased activity. However, the formation of several irreversible linkages introduces a noticeable rigidity to the protein molecule: unfolding of the enzyme is prevented and its stability rises [4,13]. The supports for this kind of enzyme immobilization must display pore diameters wider by several times than the size of the protein dimensions in order to enable good diffusion of the protein along the pore to achieve acceptable enzyme loadings, as well as high surface area.

When pore shape and size are tunable, the possibilities of studying these systems increase significantly. This is the case with siliceous ordered mesoporous materials [6,14]. These materials display uniform and regular pore systems with high interconnectivity, which facilitates not only a good enzyme diffusion to attain high enzyme loading, but also good substrate and product diffusion to decrease diffusional restrictions. Uniform pores only slightly wider than the enzyme permit high loadings of non-covalently immobilized enzyme, while a covalently-attached enzyme at the mouth of the pore would act as a plug, preventing the access of new ones, reducing the enzyme loading. Non-covalent enzyme immobilization does not require chemical modification of the protein, so the catalytic activity should not be damaged for this reason and may be better preserved. However, the favorable effect of enzyme diffusion may also lead to the unrestricted release of the enzyme, which is not possible with covalent immobilization. But when the surface of the support is coated with functional groups to provide chemical affinity with the enzyme, the situation radically changes: this affinity increases the enzyme load, and also retains the enzyme within the pore so the leaching of the reversibly linked enzyme is prevented [8]. Therefore, supports with high surface area and uniform pores with a size matching the enzyme dimensions and bearing functional groups to promote attraction, give rise to biocatalysts with high enzyme loading, retained catalytic activity and absence of enzyme leaching. These are the characteristics desired in an immobilized enzyme system.

The use of Pluronics [6] as a template for siliceous OMM formation allows for uniform pores with window/cage structures, where the large cavities or cages with wide dimensions can widely accommodate a molecule of enzyme, but the windows connecting the cages are often narrower than the enzyme dimensions. The result is a high difficulty (near impossibility) of the enzyme to diffuse through windows and a very low enzyme loading.

The harsh synthetic conditions for these siliceous ordered mesoporous materials are not compatible with enzyme activity (i.e., temperatures over 100 °C and pH below 1). It was not until milder conditions to produce these OMMs were studied and developed that in situ synthesis of the biocatalysts could be performed [9,10]. This is the fundamental idea behind the in-situ immobilization in MOFs: to build the support in the presence of the enzyme, so that a high amount of enzyme is entrapped inside the wide cages (or intercrystalline voids in aggregated nanocrystalline MOFs), and the entrapment is permanent given the narrowness of the surrounding pores, insufficient for enzyme diffusion outwards. Alternatively, in certain biomimetic strategies, the enzymes end up inside of the MOF crystals, which also avoids any leaching.

With the explosion of MOF research beginning in the late 1990s, a new horizon opened up in the field of enzyme immobilization, although this application had not started being studied until a decade later [15]. Taking advantage of the structural versatility of MOFs and the previous success of enzyme immobilization onto mesoporous materials, the first attempts to prepare biocomposite enzyme@MOF materials which were designed could only encapsulate some of the smallest proteins within crystallographic channels and/or cavities of the most porous MOFs (Ma et al. [15–19]). Thus, great efforts were made to attain MOFs containing relatively narrow mesopores to confine small proteins like cytochrome C (Cty C) [17], horseradish peroxidase (HRP) [20], or trypsin [21,22]. Crucially, these highly porous MOFs could initially only be attained with the use of very long linkers, resulting in generally unstable systems. In this context, Yang et al. proposed the idea of subjecting the MOF to ozonolysis to generate mesopores wide enough for catalase immobilization [23].

Alternatively, enzymes may become anchored onto the external surface of the MOF particles, taking advantage of the presumable chemical affinity between enzymes and MOFs in terms of the nature of functional groups, polarity, charge density distribution, etc,. However, in the absence of confinement, the adsorption of enzymes onto the external surface of MOFs is unable to prevent enzyme leaching. Therefore, some authors have proposed covalent linking, via crosslinking with glutaraldehyde [24–26] or EDC/NHS [27,28]. Also, the inclusion of new components into the composites has been often proposed, either to impart magnetic properties allowing their facile separation from reaction media [27,29], or to protect the enzyme by incorporating macromolecules like polyvinyl alcohol hydrogels [30] or by in situ formed self-assembled hybrid nanoflowers [31].

Probably, the most promising alternative is the set of strategies known as in situ, or de novo methods. As commented above, these consist of the synthesis of the MOF materials in the presence of the enzyme with the aim to entrap enzyme during the MOF formation process, either within a given MOF crystal or within the intercrystalline spaces of the aggregates formed by the fusion of the MOF nanocrystals with each other. Thus, the microporous surroundings of the MOF would prevent protein being released while allowing the diffusion of non-macromolecular substrates and products through them. However, conditions of the media for MOF synthesis are usually far from being 'enzymatically friendly'. Only when the MOF can be obtained in aqueous media under mild pH and temperature conditions can this approach be addressed [32]. Zeolitic imidazole frameworks (ZIFs) formed by the metal ions $Zn^{2+}$ or $Co^{2+}$ can be synthesized quickly and under biocompatible conditions [2], and therefore many reports have described one-pot immobilization of different enzymes, like cellulase [33] or catalase [34], among others, on ZIF-8. Apart from ZIF-8, not many MOFs can be prepared under such mild conditions, mainly due to the very low solubility of organic linkers in water. One-pot immobilization of enzymes in the MOF $NH_2$-MIL-53(Al) was patented [35] and then reported for the first time by Gascón et al. [36], based on the sustainable preparation of the carboxylate-based MOF by simple deprotonation of linkers by a base in water [37]. After this pioneering work, other enzymes have been immobilized in this material [11,38] or some other MOFs such as Fe-BTC [12,39], or CaBDC [40] which can also be prepared under mild conditions.

In order to preserve catalytic activity, macromolecules have been added in some of these one-pot systems: mixing polyvinylpyrrolidone (PVP) with Cyt C prior to the

immobilization process in ZIF-8 [32] to form a double layer to protect its activity and stability. A lignin derivative (DDVA) has also been used to co-precipitate enzymes with $Ca^{2+}$ or $Zn^{2+}$ to yield enzyme@MOM composites [41]. Additionally, $Fe_3O_4$ has successfully been added to provide particles with magnetic properties such as in the one-pot synthesis involving 2-methylimidazole and zinc acetate with lipase from *Candida rugosa* in the MOF CRL/MNP@ZIF-8 [42].

As mentioned above, there is a high affinity between MOFs and enzymes. This can be increased, for example, by making the environment of the enzyme more or less hydrophobic or hydrophilic. Thus, Liang et al. [43] described enhanced activity of catalase immobilized via one-pot synthesis in a hydrophilic environment when the linker of the MOF was 3-methyl-1,2,4-triazole (FCAT@MAF-7) compared to the hydrophobic FCAT-ZIF-8, where the enzyme undergoes inactivation. Lipase, being an enzyme which displays more activity in hydrophobic interfaces, was found to increase its activity in the hydrophobic environment created in the immobilization of lipase onto ZIF-L (AOL@PDMS-ZIF-L) and improves its stability in ZIF-8 (AOL@PDMS-ZIF-8) by the addition of PDMS (polydimethylsiloxane) to provide a hydrophobic environment [44].

Thus, it can be seen how throughout the history of enzyme immobilization, each new technique or methodology has learned and taken advantage of previous work up to the newest generations of MOF-based composites. Where previously enzyme immobilization within ordered mesoporous materials has required confined spaces, pore connectivity and chemical affinity, in situ immobilization within MOFs has provided solutions with the close retention of the enzyme in the intra- or intercrystalline spaces, and facile substrate diffusion through the porous network [45]. However, the huge structural versatility of MOFs, with hundreds of new materials discovered every year, which also present significant affinity / compatibility with so many other materials (oxides, hydrocarbons, polymers and, of course, enzymes), greatly opens the range of possibilities not only to immobilize enzymes effectively but also to rationally design an optimal habitat for the enzyme that maximizes its activity, stability and recyclability, and minimizes its leaching and inactivation. The aim of this work is to gather the progress made in the in situ/one-pot synthesis of enzyme@MOF biocatalysts thanks to the advances in the knowledge of the synthesis system and the reaction medium. It seems reasonable to start with the development of synthesis methodologies of MOFs under "enzyme friendly" conditions.

## 3. Designing MOFs Synthesis Methodologies Compatible with One-Pot Enzyme Immobilization

It is often noted that MOF materials offer a huge versatility in terms of (i) their composition (only limited by the periodic table and the known organic chemistry), (ii) their structures (thousands of different topologies are already known) or (iii) their organic functionalization (incorporated either through pre- or post-synthesis), with the wide range of applications for which these materials have been either reported or postulated [46,47]. Nevertheless, MOFs also possess other kind of versatility much less both explored and exploited: the variety in their synthesis procedures.

The stability of enzymes is relatively low, particularly their tertiary structure which gives them their biocatalytic performance. Even limited changes in temperature or acidity, the presence of alien chemical species in the media or, of course, the use of a non-aqueous solvent, could lead to the inactivation of the enzymes. Therefore, one-pot immobilization of enzymes implies that the support must be capable of being formed in the presence of enzymes under conditions that do not alter their structure/activity.

The challenge of preparing MOFs under such mild conditions is no small one. Fortunately, from the very beginning of MOF history, their preparation at room temperature has been described [48]. However, the lower quality of the resultant materials compared to their solvothermally-prepared homologues, [49,50] and the proliferation of other alternative methodologies to the solvothermal one [51] has left room-temperature approaches to be relegated for some time in the academic literature, despite the obvious sustainability benefits. In more recent times, the development of synthetic procedures capable of providing higher

quality MOFs, [37,52–58] as well as the temporal proximity of the MOFs to be applied, rekindled certain interest in these more sustainable methods.

Moreover, the materials obtained in this way, although isostructural and iso- compositional to their solvothermal counterparts, possess different physicochemical properties to those of the conventional materials. Thus, it is well-known that MOFs formed by precipitation have more structural defects; indeed, simply being nanocrystalline may make the material more defective. Furthermore, the formation of nanocrystalline MOFs results in higher external surface areas and therefore higher possibilities of creating heterojunction composites [59]. Finally, the tendency of nanocrystalline MOF crystallites to be agglomerated, or rather aggregated, in consistent and robust micron-sized particles, leads them to generate permanent intercrystalline mesoporosity with relatively uniform pore diameters [37,54]. Therefore, the so-generated MOF materials are not only much more 3E-sustainable (with 3E standing for economical, energetic, and environmental) but also their resultant properties are more adequate than those of their counterparts for certain applications such as in the direct use as catalysts and the effective immobilization of enzymes in biocomposite Enz@MOFs.

Scheme 1 arranges the general conditions for preparing MOFs via conventional (solvothermal) methodology as well as the sustainable synthetic approaches of the MOFs addressed in this review.

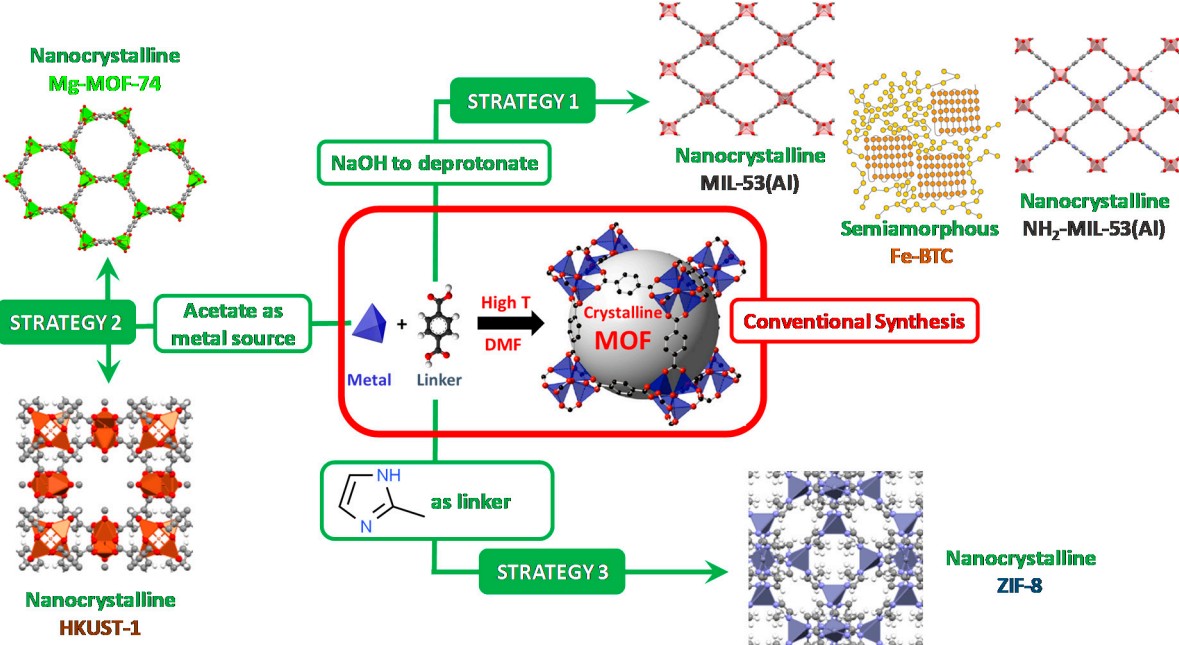

**Scheme 1.** Schematic representation of the different synthetic strategies (in green) for achieving the MOFs addressed in this work, starting from the conventional formation of a MOF (in red). The gray sphere represents the free volume within the MOF-5 cavity.

In the cases of carboxylate-based MOFs, the acidic form used as the linker source can be deprotonated by a base [37,48–50,55–57], which is essential both to favor the dissolution of the linker and also to allow the direct reaction between the metal and the carboxylate groups. Although one could imagine that the use of a base (for instance, NaOH) moves the process away from being sustainable, its role as a deprotonating agent that neutralizes the acid form of the carboxylate-based organic linker, together with its stoichiometry in the synthesis mixture, makes sure that the base cannot be found in the final reaction media. Paradoxically, the use of this base converts this system into being more sustainable, as the corrosive acid by-products generated in the solvothermal crystallization of carboxylate MOFs such as $HNO_3$, HCl or $H_2SO_4$ (depending on if the metal sources are nitrates, chlorides or sulfates, respectively) are substituted by the innocuous salts $NaNO_3$, NaCl

or Na$_2$SO$_4$ in this sustainable method [37]. This approach was used for the formation of the biocatalysts Enz@MIL-53(Al) (Section 4.1), Enz@NH$_2$-MIL-53(Al) (Section 4.1) and Enz@Fe-BTC (Section 4.2).

Alternatively, the use of carboxylates (particularly, acetates) as metal sources allows ion exchange reactions between the carboxylate-containing linkers and the acetates coordinated to the metals, to lead the formation of MOF without any additional energy input and without the addition of any chemical species as deprotonating agents or modulators [52,54,60,61]. This approach was used for the formation of the biocatalysts Enz@HKUST-1 (Section 4.3), Enz@Zn-MOF-74 (Section 4.4).

Similarly, the imidazolate-based ZIF-8 does not need any of these stimuli as the simple contact of metal and linker readily leads to the formation of the MOF material [53,62–66]. The ease of formation of ZIF-8 is promoted by the high solubility of the 2-methylimidazole linker in water, allowing for spontaneous formation of ZIF-8 at room temperature. This approach has been used for the formation of a large number of Enz@ZIF-8 biocatalysts (Section 4.5).

It must be noted that, unlike the syntheses outlined in Scheme 1, the synthesis of the in-situ biocatalysts Enz@MOF implies that the enzyme itself is present in the synthesis media of the MOF support. Therefore, it could potentially alter the chemistry of the synthesis media as well as the formation of the MOF, especially in situations whereby (i) there is significant interactions between the enzymes and the MOF [67,68] and (ii) the enzyme molecules contain carboxylate groups similar to those of some the above-mentioned linkers that form MOFs by bonding with metallic clusters. As a consequence, the presence of enzymes could change the formation kinetics, the appearance of impurities, the defects, the crystal size, the intercrystalline mesoporosity, etc., of the resultant MOF-based material.

## 4. One-Pot MOF-Based Biocatalysts

Scheme 2 shows a comparison between the one-pot and the post-synthesis procedure for enzyme immobilizations onto MOF-based supports, as well as the advantages and drawbacks of both methodologies. Although each enzyme@MOF biocatalyst should be individually studied in detail, in general terms, the one-step methods offer greater advantages.

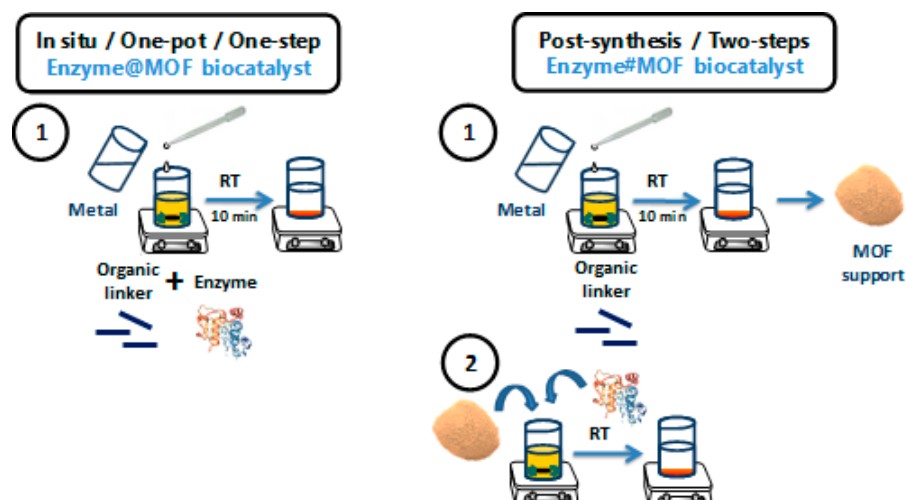

**Scheme 2.** Schemes of the one-step in situ methodology (**left**), and the two-step or post-synthesis methodology (**right**) for the immobilization of enzymes onto MOF-based supports prepared under mild conditions. This figure has been inspired by ref. [68]. Of course, the first step on the post-synthesis methodology, that is, the preparation of the enzyme-free MOF, could be carried out under conventional conditions, making this process potentially more laborious and damaging to the environment.

One-pot enzyme immobilization has previously been described for a variety of MOFs, as indicated in Scheme 1. However, any given strategy (for instance, the deprotonation approach in Scheme 1) is dependent on the specific features of the MOF supports, such as the nature of the linker (carboxylates, imidazolates, etc.) and functionalization, their intrinsic intercrystalline mesoporosity, their crystallite size, the pH of their synthetic media, the nature of the metal, etc. These factors strongly influence the compatibility of the specific biocomposite, with subsequent effects on the enzyme catalytic activity, the affinity for enzymes, or the immobilization efficiency. For that reason, this section is divided into different Sections according to the type of MOF material used as support, starting from the carboxylate-based MOF and finishing with the imidazolate-based ZIFs, with ZIF-8 being the most widely used sustainable MOF support for enzymes. Table 2 summarizes the strategies, the MOF supports and the enzymes forming the biocomposite Enzyme@MOFs discussed in this review.

**Table 2.** Summary of the strategies, MOF supports, enzymes and biocomposite Enzyme@MOFs covered in this work. The number used for denoting the different strategies is in accordance with those used in Scheme 1.

| Strategy | MOF | Enzyme | References |
|:---:|:---:|:---:|:---:|
| 1 | Fe-BTC | Laccase | [39] |
| | | Lipase | [12,39] |
| | | Alcohol dehydrogenase (ADH) | [12,69] |
| | | Glucose oxidase (GOx) | [12,70] |
| | | Peroxidase (POx) | [70] |
| 1 | NH$_2$-MIL-53(Al) | β-Glucosidase (β-Glu) | [36] |
| | | Laccase | [38] |
| | | Lipase | [11] |
| 2 | HKUST-1 | Glucose oxidase (Gox) | [71,72] |
| | | Horseradish peroxidase (HRP) | [71,72] |
| | | Laccase | [61] |
| | | Urease | [72] |
| 2 | Mg-MOF-74 | β-Glucosidase (β-Glu) | [36] |
| 3 | ZIF-8 | Alcohol oxidase (AOx) | [73] |
| | | Carbonic anhydrase (CA) | [74] |
| | | Catalase | [34,43,75–78] |
| | | Cytochrome C (Cty C) | [32,79] |
| | | Glucose oxidase (GOx) | [72,78–80] |
| | | Horseradish peroxidase (HRP) | [72,73,77,80,81] |
| | | Laccase | [82,83] |
| | | Lipase | [77,78,81,84,85] |
| | | Lysozyme | |
| | | Pyrroloquinoline quinone Glucose dehydrogenase (PQQ-GDH) | [81] |
| | | Ribonuclease A | [81] |
| | | Trypsin | [81] |
| | | Urease | [81] |
| | | β-Galactosidase | [80,86,87] |
| 3 | ZIF-90 | Catalase | [43,75,88] |
| | | Superoxide dismutase | [89] |
| 3 | Amorphous-ZIF | Catalase | [78] |
| | | Glucose oxidase (GOx) | [78] |
| | | Lipase | [78] |
| 3 | ZIF-L | Carbonic anhydrase (CA) | [90] |
| 3 [a] | MAF-7 | Catalase | [43] |

[a] Needs a deprotonation agent.

### 4.1. MIL-53(Al) and NH₂-MIL-53(Al) as a Tandem to Compare Post-Synthesis and In Situ Enzyme Immobilization

MIL-53(Al) is one of the best-known flexible MOFs. Despite being formed of rigid linkers and relatively strong carboxylate-aluminum bonds, it possesses the ability to reversibly adopt different structures without the formation or breaking of a single bond, in response to external stimuli (temperature, pressure, hydration, etc.) [91]. The room-temperature synthesis of X-MIL-53(Al) materials (X = none, –NH₂, –NO₂) in water has only been described when assisted by a base, either inorganic (NaOH or NH₄OH, which are strong and medium strength bases, respectively) or organic (such as triethylamine) [36,37].

Despite both MIL-53(Al) and NH₂-MIL-53(Al) being successfully prepared under the same conditions, there are differences in the kinetics of their formation and in their intercrystalline mesoporosity [37], which is believed to be essential to effectively trap enzymes in the resultant biocatalysts. Moreover, the intrinsic tendency of being formed by this sustainable methodology is also different. MIL-53(Al) needs synthesis times as long as four days to complete its crystallization, whereas NH₂-MIL-53(Al) scarcely needs a few min/h [11,37,38]. Presumably, the difference in kinetics, which is due to the different solubility of terephthalic acid and 2-aminoterephthalic acid, could also lead to the difference in crystal size and also in intercrystalline mesoporosity, with pore size distributions centered at 35 and 5 nm, respectively. The mesopores of MIL-53(Al) are in fact too large for an effective confinement of enzymes (whose globular diameter is usually within the range of 3–10 nm) by the post-synthesis methodology (Figure 1) and, in addition, the absence of NH₂-groups entails a serious disadvantage in the immobilization of enzymes like laccase versus its counterpart NH₂-MIL-53(Al) [36].

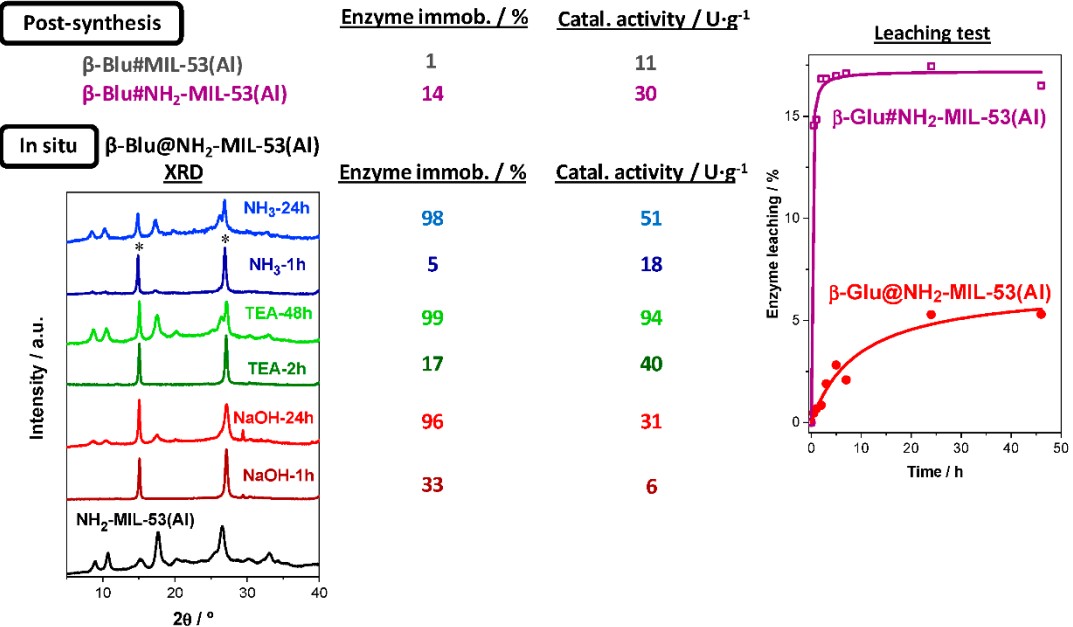

**Figure 1.** Enzyme immobilization (as a percentage) and catalytic activity (in activity units per g of biocatalyst in the hydrolysis of *para*-nitrophenyl-β-D-glucopyranoside) for biocatalysts prepared by immobilizing the enzyme beta-glucosidase (β-Glu) on NH₂-MIL-53(Al) (and on MIL-53(Al) for a particular case) via post-synthetic and in situ methodologies. The XRD patterns of the in situ prepared biocatalysts β-Glu@NH₂-MIL-53(Al) using different deprotonating agents (NaOH, TEA and NH₄OH) are shown (on the bottom left); both asterisks mark the two sharp and intense peaks attributed to H₂BDC. Kinetics of enzyme leaching for the biocatalysts prepared through post-synthesis (β-Glu#NH₂-MIL-53(Al), purple) and in situ (β-Glu@NH₂-MIL-53(Al)-NaOH-24, red) methodologies are also plotted (on the right). Data extracted from [36].

The XRD patterns of Figure 1 make clear the influence of enzymatic extract on the chemistry of NH₂-MIL-53(Al). Whereas this MOF is purely and quickly formed either in the

absence of any enzyme [37] or in the presence of different laccase extracts [11,38], the beta-glucosidase extract retards considerably the formation of the MOF, which does not become completely pure (with terephthalic acid also present) even after 48 h. Three more important conclusions can be extracted from Figure 1: (i) the nature of the deprotonating agent (NaOH, NH$_4$OH or triethylamine (TEA)) influences both the NH$_2$-MIL-53(Al)/H$_2$BDC phase ratios and the activity of the resultant beta-glucosidase@NH$_2$-MIL-53(Al) (specific activity per mg of enzyme is much lower for samples prepared with NaOH as the deprotonating agent); (ii) enzyme β-glucosidase is retained on the MOF phase and not on the H$_2$BDC phase; all enzyme was retained even in samples with relatively poorly crystalline NH$_2$-MIL-53(Al) (see samples NaOH-24 h); and (iii) the in situ methodology surpasses the post-synthesis methodology in key aspects such as the effectiveness of enzyme immobilization, catalytic activity, and in minimizing the enzyme leaching.

　　　Figure 2a,b show the effect of both the pH and the temperature of the reaction medium on the catalytic activity of the biocatalyst laccase@NH$_2$-MIL-53 with respect to the free enzyme. Remarkably, the immobilized laccase is systematically more active (at any pH or T) than the free laccase and, at the same time, its activity is less sensitive to the change in reaction conditions. In other words, the MOF-based support not only encourages greater activity for the enzyme laccase, but also stabilizes it. Such stabilization is also clear from Figure 2c, which shows the kinetics of the catalytic activity at 60 °C.

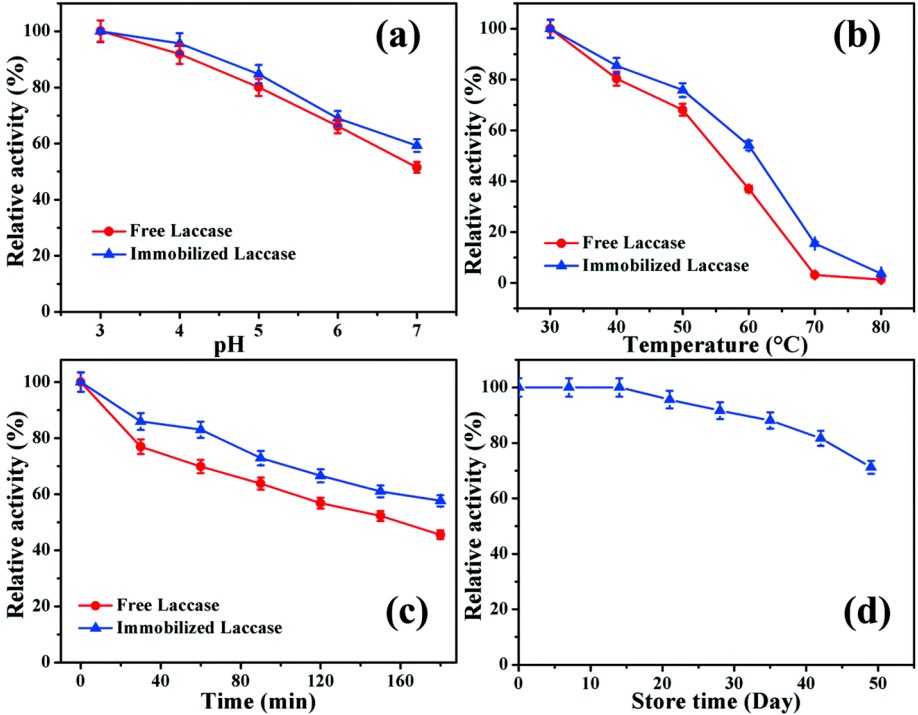

**Figure 2.** Catalytic activity of free laccase (red) and laccase@NH$_2$-MIL-53(Al) (blue) in the oxidation of ABTS at different pHs (**a**), at different temperatures (**b**); thermal stabilities of free and immobilized laccase at 60 °C (**c**) and storage stability of immobilized laccase (**d**). Reproduced from [38] with permission from the Centre National de la Recherche Scientifique (CNRS) and the Royal Society of Chemistry.

　　　In summary, although there are only a few studies on enzyme immobilization in (NH$_2$)-MIL-53(Al), these provide decisive conclusions for a better understanding of the one-pot enzyme@MOF systems: (i) beyond the sustainability of the prepared biocatalyst, the benefits of the in situ methodology generally outweigh their drawbacks; (ii) in the in situ methodology, enzymes are selectively immobilized on their own MOF and not on other related impurities like the protonated linker (terephthalic acid); (iii) enzymatic

extracts, far from being mere spectators, modifies the chemistry of the synthetic media and therefore the properties of the resultant enzyme-MOF biocomposites; and (iv) the support NH$_2$-MIL-53(Al) provides thermal and temporal stability to the immobilized enzyme (at least for laccase), which maintains or even slightly surpasses the activity given by the free-enzyme counterpart.

### 4.2. Semiamorphous Fe-BTC

Although MOFs are quite often defined as crystalline materials, they can also be amorphous or semiamorphous, according to IUPAC [92]. Indeed, semiamorphous Fe-BTC, commercialized as Basolite F300, is one of the most widely tested as direct heterogeneous catalysts even before its direct synthesis was described [93]. This material is of unknown structure due to its semiamorphous nature but it is well known that it is highly related to MIL-100(Fe), as both have the same metal, the same linker, similar thermal stability and one of their mesocavities in common [56,57,94,95]. Moreover, it is presumed that the semiamorphous character of the former, as well as the precipitation methodology used for its preparation, gives this material a higher amount of defects than MIL-100(Fe), which could have a key role in the immobilization of enzymes.

Among the known MOFs, use of semiamorphous Fe-BTC as a protein support has been relatively prolific [1,12,39,70,81]. As the in situ methodology is not sensitive to the size of the support pores, nor does the size of the enzyme limit the preparation of an active biocatalyst, this approach has been extended to a vast range of enzymes with different molecular weights [12,39], including laccase (LAC), lipase (LIP), alcohol dehydrogenase, e (ADH), glucose oxidase (GOx) and halophilic HvADH2, among others. As a particular example, the resultant activity of the CALB Lipase@Fe-BTC biocatalyst was retained up to 97 % with respect to the free enzyme [39]. In particular, this Fe-BTC enzyme support improves upon the benefits given by some other MOF-based supports such as, NH$_2$-MIL-53(Al) [11,36].

The formation of Fe-BTC is very rapid (in less than 10 min), it can be synthesized under mild conditions (in water solution, moderate pH and at room temperature) and with close to total encapsulation of the enzyme available in the reaction media [12,39].

Unlike the immobilization of GOx and LIP, the immobilized ADH from *Saccharomyces cerevisiae* retained only 6% of the activity in comparison with the free enzyme present initially in the solution. Since this enzyme requires the addition of the bulky cofactor NAD$^+$, with consequent diffusion restrictions, co-immobilization of the cofactor and ADH demonstrated an enhanced performance in both reusability and catalytic activity [12].

Other reports describes the in situ immobilization of the halophilic HvADH2 in the Fe-BTC MOF [69]. Enzyme specificity, stability and tolerance to organic solvents were systematically studied. With this in situ approach, unlike other immobilization methods, the biocatalyst resulted in increased stability over a wider range of pH and temperature with retention of activity upon reuse of up to 4 cycles [69]. Electrostatic interactions between the halophilic enzyme and the Fe-BTC MOF might explain the enhancement in activity and decrease in halophilicity of the immobilized enzyme. The catalytic activity of the immobilized enzyme was studied in solvent mixtures with the highest retention of activity in methanol and acetonitrile. This approach demonstrates that this immobilization method can be extended to hyperhalophilic enzymes with enhancements in activity and stability.

The process used to immobilize enzymes in the Fe-BTC MOF material [12] can be extended under different operating conditions to a broad range of enzymes, improving their properties. The interactions between the material and the enzyme provide a favorable microenvironment, broadening the operational conditions. Encapsulation of the enzyme resulted in an increase of optimal work temperature (i.e., from 50 °C to 60 °C), a broader range of working pH, a decrease in the requirement for high concentrations of salt, good storage stability and retention of activity in organic media such as Dimethyl sulfoxide (DMSO) and acetonitrile which is not achievable with the free enzyme. Fe-BTC provides enough stability to retain the enzyme activity and biocatalyst performance (i.e., See Figure 3,

the relative catalytic activity of GOx, LIP and ADH is maintained during 5 reaction cycles). Future work could be devoted to testing the performance of other enzymes integrated in a cascade system [70] or even other classes of macromolecules [96,97] and to verify the feasibility of a continuous processes by using a continuous fixed-bed reactor or on larger-scale production.

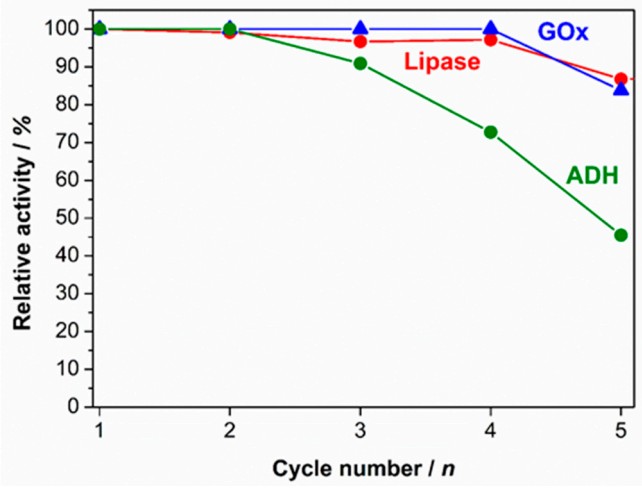

**Figure 3.** Relative catalytic activity during 5 reaction cycles with the biocatalysts prepared in one step, formed by the semicrystalline Fe-BTC material as a support and the enzymes glucose oxidase@GOx (blue), lipase@Lip (red), or alcohol dehydrogenase@ADH (green) as the active species. The catalytic activity was measured for an optimized reaction for each enzyme for enzymatic activity assays) and is compared to the catalytic activity of each biocatalyst in its first reaction cycle, which was taken to be 100 %. Reproduced with permission from [12].Copyright Wiley-VCH Verlag GmbH & Co. KGaA.

### 4.3. HKUST-1

HKUST-1 (Cu-BTC) was one of the first reported MOFs [98] and its preparation at room temperature has been widely studied. It can be prepared under mild conditions and has been also used for in situ immobilization of enzymes. Unlike Fe-BTC, Cu-BTC is perfectly crystalline [99]. This material forms face centered-cubic crystals that contain an intersecting 3D system of large square-shaped pores ($9 \times 9$ Å), which are of insufficient size to harbor enzymes.

However, HKUST-1, which can be prepared with many different morphologies, can form layers where enzymes can be entrapped. For example, very recently Zhang et al. [61] achieved enhanced activity and improved stabilization of laccase into the layer formed by HKUST-1 through a biomimetic mineralization process. The authors propose that there is a coordination between the amide groups in the laccase surface and the copper ion which act as nucleation points to start the biomineralization process. The procedure to obtain the laccase@HKUST-1 composites takes place in aqueous media and mild conditions, by mixing cupric acetate monohydrate solution, that also contains laccase, with BTC solution in an acetate buffer, at 30 °C. Similarly, Chen et al. [72] developed a rapid method for encapsulation of proteins via a biomimetic strategy using ZIF-8 as a support, although this versatile methodology can be extended to other MOFs, including HKUST-1. Chen et al. [71] prepared a magnetic HKUST-1 metal organic framework, also under mild conditions, by alternating layers of the MOF and the magnetic $Fe_3O_4$ nanoparticles. The enzymes were either encapsulated into the HKUST-1 inner layers, or immobilized at the HKUST-1 outer shell, or randomly distributed within the two MOF layers.

### 4.4. Mg-MOF-74 Prepared in Non-Aqueous Systems

Amongst all MOF families, the MOF-74 family is known to be one of the most versatile in their composition [100–102] but also in their preparation. Thus, the wide temperature

range (between −78 and 120 °C) and the variety in possible solvents (at least, five have been described) in the synthesis of MOF-74 is unknown for any other MOF. In addition, the preparation of the MOF-74 materials, starting from metal acetates as metal precursors, in either methanol [103] or in DMF (N,N-dimethylformamide) [54] as solvents, and at room temperature generates the MOF materials with the smallest crystallites / domains ever reported. These nanosized MOF-74 crystallites possesses intercrystalline mesopores of quite uniform diameter and external surface area as high as the microporous area [54].

These exceptional properties for enzyme supports encouraged us to test the performance of this MOF-74 material in the one-pot immobilization of the enzyme beta-glucosidase, in spite of this synthesis being carried out in a solvent as enzymatically and environmentally-unfriendly as DMF (Figure 4) [54].

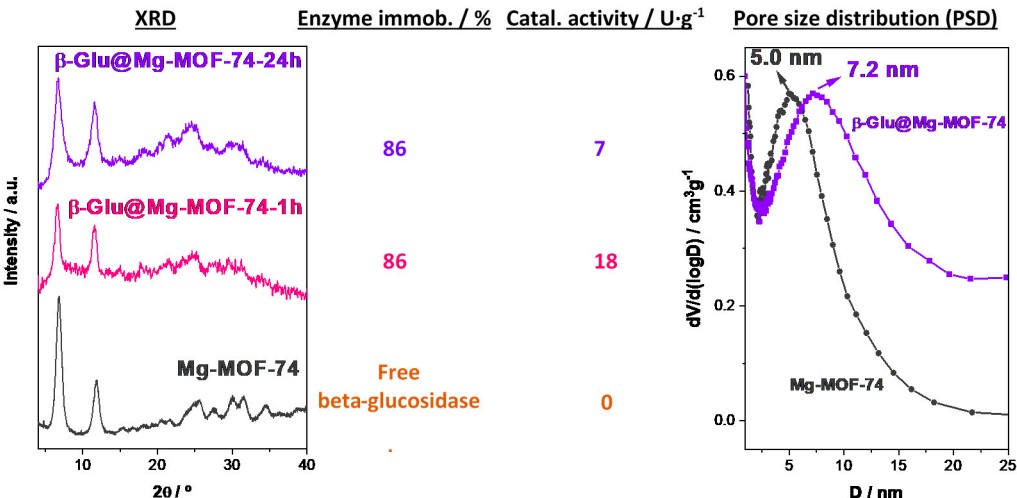

**Figure 4.** XRD patterns, enzyme immobilization (in percentage), catalytic activity (in activity units per g of biocatalyst in the hydrolysis of *para*-nitrophenyl-β-D-glucopyranoside) and PSD curves of the biocatalysts prepared by in situ immobilization of the enzyme β-glucosidase (β-Glu) on Mg-MOF-74 in DMF after 1 or 24 h. The activity of the free enzyme in contact with DMF and the maximum of the PSD peaks are indicated. Data extracted from ref. [36].

The XRD patterns of the biocatalysts β-Glu@Mg-MOF-74 show that, unlike in the NH$_2$-MIL-53(Al) system, this enzyme does not change the crystallized phases, although the low intensity and resolution of the diffractograms of the samples containing enzyme indicate the active role of beta-glucosidase in the formation of the MOF. Indeed, the enzyme, which is bigger than the intrinsic average intercrystalline mesoporosity (5 nm), enlarges the pore size distribution of the mesoporous channels (up to 7.2 nm), strongly suggesting that the enzyme molecules, unlike what happens in the biomimetic pathway, are found in the intercrystalline mesopores.

This study is also unique due to the enzyme immobilization being carried out in DMF rather than aqueous solution. It must be noted that the free beta-glucosidase becomes immediately inactive as soon as it comes into contact with the solvent DMF. However, the Mg-MOF-74 support is able to stabilize the enzyme in this hostile synthetic medium, as the biocatalyst β-Glu@Mg-MOF-74 continues being active after 24 h submerged in DMF (Figure 4) [36]. It is evident that Mg-MOF-74 not only supports the enzyme, but also contributes to keeping the stability of the enzyme against adverse external stimuli.

### 4.5. ZIFs, the Most Widely Studied MOFs as a One-Pot Support for Biocomposites

All the MOF materials discussed so far are carboxylate-based MOFs (Table 1). By contrast, zeolitic imidazole frameworks (ZIFs) [104] form a vast family of MOF materials differentiated from carboxylate-based MOFs in some key properties: (i) the metal node is just a metallic ion (normally the divalent ions Zn$^{2+}$ or Co$^{2+}$) and not a metal cluster; (ii) the

linker is based on 5-atom aromatic rings with at least 2 nitrogen atoms (imidazolates, tria-zolates or tetrazolates); (iii) the angle formed between two consecutives metals separated by the linker (which is quite close to 145 °, practically equal to the average T–O–T angle found in zeolites) means that they have zeolitic-like topologies, although some structures described for ZIFs do not have a known zeolitic homologue so far.

Like the carboxylate-based MOFs, all ZIFs can be synthesized using solvothermal methods [104]. However, they can be also prepared by environmentally friendly ap-proaches (Scheme 1). Indeed, ZIF-8 is by far the most studied MOF for one-pot enzyme immobilization taking advantage of the fact that it can be synthesized using biocompatible conditions that appear to be well tolerated by several enzymes (Table 1) [1]. Unlike the aromatic linkers containing carboxylates, particularly terephthalic acid, the corresponding acid form of the ZIF linkers are quite soluble in water, which, combined with the fact that a metal cluster does not need to be formed, greatly facilitates the spontaneous formation of the ZIF in aqueous solution by simply mixing linker and metal precursor.

The first approaches in the preparation of a MOF-supported biocatalyst using ZIFs employed a coprecipitation method in which the enzymes were introduced into a solution containing the ZIF precursors with a capping agent (i.e., polyvinylpyrrolidone, PVP), in order to form a double layer to protect enzyme activity [32,74,75]. Meanwhile, another approach, called 'biomimetic mineralization', was carried out in the absence of capping agents. In this methodology, the biomacromolecule (enzyme in this case) induces the growth of the MOF in water, in such a way that the macromolecule ends up embedded inside a MOF crystal [34,76,81,105,106]. Liang et al. assessed the relative efficacy of each approach by comparing the thermal stability of encapsulated urease (Figure 5) [107]. They determined that both approaches exhibited comparable encapsulation efficiencies suggest-ing that, in aqueous solutions, PVP does not play a role in enhancing biomacromolecule loading and stability.

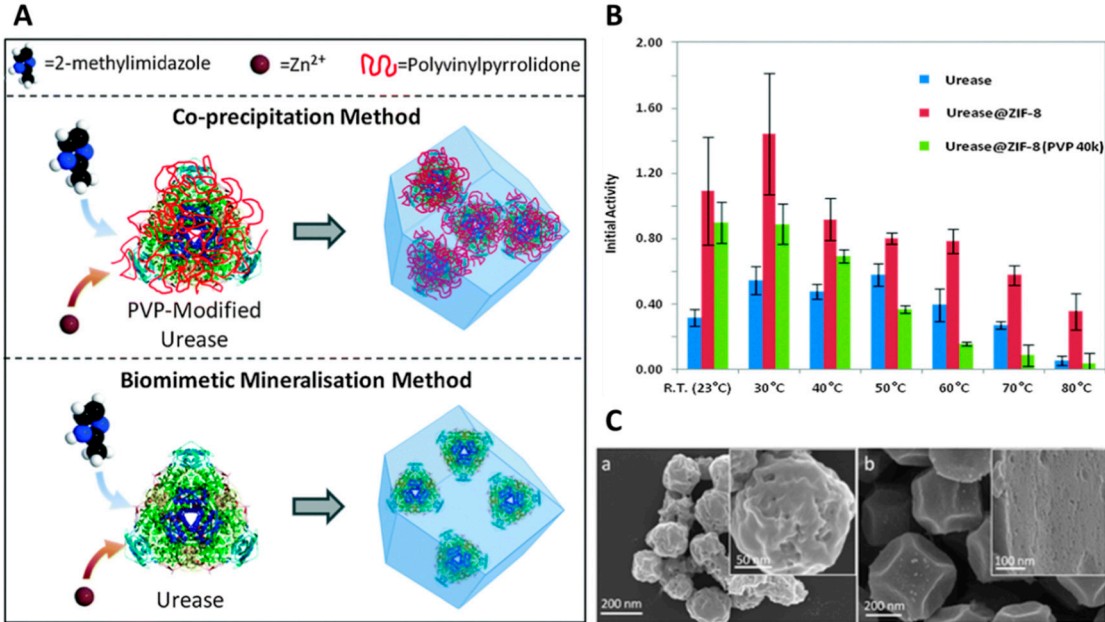

**Figure 5.** (**A**) Urease@ZIF-8 prepared with (top) and without (down) PVP as an additive. (**B**) Initial activity at different reaction temperatures. (**C**) SEM image of urease@ZIF-8 prepared with (**C**-**a**) and without (**C**-**b**) PVP as an additive. Adapted with permission from [107]. Copyright 2016 Royal Society of Chemistry.

The role of proteins in the material formation process has been the subject of contro-versy. Some authors reported a biomimetic mineralization approach that utilizes biomacro-molecules (proteins) as crystallization and directing agents for controlling crystal morphol-ogy of ZIF-8 [77,79,81,106]. However, Cui et al. demonstrated that the crystal morphology

of ZIFs is primarily dependent on the concentrations of 2-methylimidazole and $Zn^{2+}$ ions instead of the biomacromolecules (proteins) [76], as other authors had confirmed [81,106]. Other works suggest that the $Zn^{2+}$:linker:enzyme ratios and the total precursor concentration are important parameters that determine the phase and the morphology of the obtained enzyme@ZIF composite [73,78,90,108,109]. Although the relationship between the ZIF phase and its ability to protect the activity of the encapsulated enzyme has not been demonstrated, it is likely that the different physicochemical properties of each phase determine the properties of the biocomposite. Thus, a fast crystallization process favors high catalytic activity retention [110] and leads to particles of smaller size. For example, in the lipase from *Candida antarctica* B immobilized in ZIF-8 (CaLB@ZIF-8), the small particles are afforded at low Zn:linker ratios and indeed keep higher catalytic activity [85]. Furthermore, the glucose oxidase (GOx) enzyme encapsulated in amorphous ZIF was 20 times more active than that encapsulated in crystalline ZIF-8 [78].

It is known that some enzymes have good affinity for hydrophobic surfaces [111]. Therefore, adsorption of enzymes onto these surfaces can cause changes in their three-dimensional structure that result in the loss of activity. Liang et al. studied the influence of the hydrophobicity of different ZIFs on catalase activity (Figure 6) [43]. They prepared catalase@ZIF composites by varying the organic linker: hydrophobic ZIF-8 (linker: 2-methylimidazolate), hydrophilic MAF-7 (linker: 3-methyl-1,2,4-triazolate), and ZIF-90 (linker: 2 -imidazolate carboxaldehyde). It was found that the activity of the enzyme encapsulated in the more hydrophilic MAF-7 retained higher activity than in the more hydrophobic compound ZIF-8. Furthermore, the ability to protect the enzyme from hostile conditions was enhanced for the more hydrophilic matrixes.

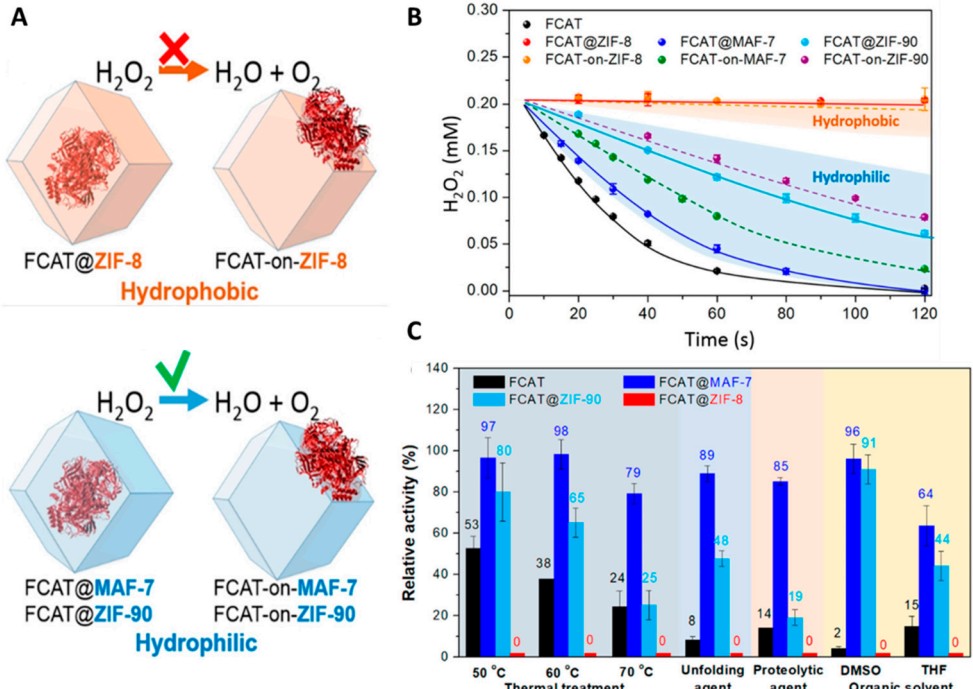

**Figure 6.** (**A**) MOFs with varying degrees of hydrophobicity on catalase immobilization on MAF-7, ZIF-90, and ZIF-8 by post synthesis or one-pot methods. (**B**) Activity data for catalase immobilization on MAF-7, ZIF-90, and ZIF-8 by post synthesis or one-pot methods compared with the free enzyme. (**C**) Activity of biocatalysts after thermal treatment, in the presence of chaotropic agent (urea), in the presence of proteolytic agent (4 mg mL$^{-1}$ protease, 2 h), or after exposure to organic solvent (DMSO or tetrahydrofuran (THF), 2 h). Figure adapted with permission from [43]. Copyright 2019 American Chemical Society.

## 5. Conclusions

Sustainability is undoubtedly the main challenge for current advances in chemical processes. Enzymatic catalysis fulfils most requirements of green chemistry regarding reaction conditions, but the necessity of working with immobilized enzymes as a result of the lability and solubility of these proteins threatens to become a new source of environmentally unfriendly processes. Therefore, the development of sustainable methods for the immobilization of enzyme has gained significant attention and some of these methods are reviewed herein. The advantages of the use of MOFs for in situ enzyme immobilization (low cost, leaching prevention of the entrapped enzyme, and sufficient substrate and product diffusion) can be exploited, but the sustainable synthesis of these biocatalysts is not without challenges. Some MOFs, like ZIF-8 and HKUST-1, do not require harsh conditions in their preparation and therefore their synthesis in the presence of enzymes produces one-step biocatalysts in non-polluting conditions. In other cases, like Fe-BTC or $NH_2$-MIL-53(Al), modification of their respective synthetic procedures may be required in order for synthesis to occur in the presence of enzymes. These new conditions are based on the preservation of catalytic activity of the enzymes under sustainable conditions, namely aqueous medium, mild pH, and room temperature. A summary of the field is offered here showing how these systems may offer catalytic activity preservation and/or enzyme stabilization depending also on secondary factors such as the kind of interaction between the enzyme and the organic linker. This revision is meant as a starting point to the further studies of mild-condition synthesis of new Enzyme@MOF catalysts.

With the potential for in situ immobilization in/on MOFs being presented in this manuscript, some issues about future perspectives of these materials and their applications can now be reflected and advised upon. First of all, it is expected that the evolution to an increasingly sustainable world, particularly in chemical processes, will make these methodologies progressively gain ground in the general context of the immobilization of enzymes and MOFs. Secondly, it is worth noting that both the particular MOF support and its synthetic conditions must be optimized according to the nature of the immobilized enzyme, and the intended use of the resultant biocomposite; aspects as relevant as toxicity of metals and linkers, synthetic pH, the nature of deprotonating agents (if any), and the role of the enzyme (biomimetic, intercrystalline mesopore swelling, etc.) could decisively determine the scope of the enzyme@MOF application. Thirdly, the wider scientific community should take advantage of the well-known catalytic potential of MOFs to lead one-pot biocomposite enzyme@MOFs to where MOFs are not mere supports but become active participants that favor chain reactions, provide synergistic effects with the enzymes, or encourage shape selectivity (before, after or during the enzymatic reaction) for further performance. Finally, increasingly powerful computational calculations and characterization techniques should lead to a more exhaustive knowledge of the exact location of the enzymes (inter or intracrystalline) and of the interactions (even at the atomic level) at play in the MOF support, as well as its influence on the catalytic activity of the resulting MOF leading to an acceleration of development in this field.

**Author Contributions:** All four authors (M.A.M., V.G.-P., M.S.-S. and R.M.B.) have actively participated in the design, writing and revision of this article. R.M.B. and M.S.-S. gave the initial focus to the article and organized the final content. M.A.M. was in charge of the bibliographic searches and everything related to the final format. All authors have read and agreed to the published version of the manuscript.

**Funding:** This research was funded by CSIC (2019AEP076). M.A.M. acknowledges Spanish MINECO for PhD student contract BES-2017-082077. English-writing style. The Government of Ireland Postdoctoral Fellowship-2015 GOIPD/2015/287 is gratefully acknowledged by V.G.-P.

**Acknowledgments:** The authors want to thank Ramiro Martinez (Novozymes) for his kind gift of the enzymes. The authors thank the valuable help of Yassin H. Andaloussi for his suggestions and advices with English-writing style.

**Conflicts of Interest:** The authors declare no conflict of interest.

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
