# Peer review of "Sustainable One-Pot Immobilization of Enzymes in/on Metal-Organic Framework Materials"

_catalysts, doi:10.3390/catal11081002_

Round 1

Reviewer 1 Report

This is an interesting study, and the paper is well written and structured. However, I found following issues in the paper.

  1. Line 440-441 – the author must use ‘has’ instead of ‘have’ for a singular pronoun.
  2. Line 449- spelling of copper is wrong.  
  3. Either use in situ or in-situ.
  4. Line 91- a unit of temperature is wrong.
  5. Line 170- remove the space between in situ and slash.
  6. Line 137- et al is incorrect.
  7. Line 435- Reproduced with permission. [34]. Abbreviation of reference is missing.
  8. Either use Ref. or ref.
  9. Correct the references 14, 24, and 41.
  10. Write the name of all authors in ref. 14, 16, 38, 67, 70, 77, 84, 97, 98

Author must fix above mistakes prior to acceptance.

Author Response

 Reply#1 to Reviewer#1 of the manuscript catalysts-1343026

Journal: Catalysts

Manuscript ID: catalysts-1343026

Title: ‘Sustainable one-pot immobilization of enzymes on/in MOFs’

Authors: M. Asunción Molina, Victoria Gascón-Pérez, Manuel Sánchez-Sánchez and Rosa M. Blanco

First of all, we would like to thank Reviewer#1 for his/her interest in reading our manuscript and for the effort made in the evaluation, which undoubtedly has helped us to improve the original version.

Please find below detailed replies (in blue text) to any comment/suggestion made by Reviewer#1. The changes in the new version of the manuscript have been introduced with the ‘Track Changes’ Tool of Word.

Reviewer #1: This is an interesting study, and the paper is well written and structured. However, I found following issues in the paper.

Thanks to Reviewer#1 for so encouraging opinion.

  1. Line 440-441 – the author must use ‘has’ instead of ‘have’ for a singular pronoun.

It has been changed.

  1. Line 449- spelling of copper is wrong.  

It has been corrected.

  1. Either use in situ or in-situ.

The term ‘in situ’ has been homogenized along the manuscript, and ‘in-situ’ has been avoided.

  1. Line 91- a unit of temperature is wrong.

We have modified that in the mentioned line and also along the manuscript.

  1. Line 170- remove the space between in situ and slash.

It has been removed.

  1. Line 137- et al is incorrect.

‘et al’ has been changed by ‘et al.’

  1. Line 435- Reproduced with permission. [34]. Abbreviation of reference is missing.

It has been now included.

  1. Either use Ref. or ref.

We have changed ‘Ref.’ by ‘ref.’ along the whole manuscript.

  1. Correct the references 14, 24, and 41.

They have been corrected in the new version of the manuscript.

  1. Write the name of all authors in ref. 14, 16, 38, 67, 70, 77, 84, 97, 98

Sorry for that. It was a problem with the reference editor software we used. It has been now corrected.

Author must fix above mistakes prior to acceptance.

Hopefully, the changes introduced in the manuscript, together with this detailed reply to any Reviewer’s comments, will make it of enough quality for being accepted in Catalysts.

Reviewer 2 Report

Authors have addressed a very important topic of today- sustainability! and review is well written. Please address few minor corrections below-

  1. Please provide citations for table 1.
  2. Page 2, line 60- some important citations can be added describing rise in activity post covalent modification of enzymes, for eg- doi.org/10.1021/ACSCATAL.5B00958
  3. Page 3, line 113, please clean up the typo error- ...,"talking" advantage...
  4. Page 3, line 117 lot of good examples can be found on this and can cited here. 
  5. In scheme 1, in the red rectangle, what does the grey ball indicate? please specify. 
  6. Since this is a review, authors may be missing out on citing at appropriate locations. More citations and references will give readers more avenues to cross-reference and be more impactful. For eg- page 6 can have more references. 
  7. In scheme 2, very specific enzyme and linker structures are used. Can authors specify the names? If they are referring the process to be more generic, then, more generic schemes/symbols should be used.
  8. Table 2 is really well formed. 
  9. Figure 2 (d) does not have comparison with free enzyme. Can authors comment on this.

Author Response

Reply#1 to Reviewer#2 of the manuscript catalysts-1343026

Journal: Catalysts

Manuscript ID: catalysts-1343026

Title: ‘Sustainable one-pot immobilization of enzymes on/in MOFs’

Authors: M. Asunción Molina, Victoria Gascón-Pérez, Manuel Sánchez-Sánchez and Rosa M. Blanco

We are very grateful to Reviewer#2 for his/her carefully reading our manuscript and evaluating it in a constructive way.

Please find below detailed replies (in blue text) to any of his/her comment/suggestions. The changes in the new version of the manuscript have been introduced with the ‘Track Changes’ Tool of Word.

Reviewer #2: Authors have addressed a very important topic of today- sustainability! and review is well written. Please address few minor corrections below-

  1. Please provide citations for table 1.

We have added at least one reference per column (that is, per each enzyme immobilization strategy), including a new one (ref 7). We hope it is enough, at least as bibliographic starting point, for readers interested in any of these historical and conventional approaches.

  1. Page 2, line 60- some important citations can be added describing rise in activity post covalent modification of enzymes, for eg- doi.org/10.1021/ACSCATAL.5B00958

We agree about the convenience of adding a reference at that point, apart from the one already gave by us. The recommended reference seems to us very appropriate.

  1. Page 3, line 113, please clean up the typo error- ...,"talking" advantage...

It has been corrected.

  1. Page 3, line 117 lot of good examples can be found on this and can cited here.

Some new references have been added in the new version of the manuscript.

  1. In scheme 1, in the red rectangle, what does the grey ball indicate? please specify.

Well, it is quite common in the graphical representations of MOF structures to highlight the void space delimited by the ordered arrangement of organic linkers and metal clusters. Accordingly, such grey ball represents the maximum free volume within a MOF-5 cube. Although it is well-known by all MOF community, the meaning of such ball has been clarified in the Scheme 1 caption. 

  1. Since this is a review, authors may be missing out on citing at appropriate locations. More citations and references will give readers more avenues to cross-reference and be more impactful. For eg- page 6 can have more references.

This review focuses on a particular approach for preparing biocomposites enzyme@MOF. Indeed, although several thousands of MOFs are known, just a few of them (ZIF-8, NH2-MIL-53(Al), HKUST-1, Fe-BTC) have been successfully used so far with this purpose. As a consequence, the number of papers directly related to the goal of the review is relatively reduced amongst the reported enzyme@MOF. Therefore, we think that 100 references (111 in the new version) for a review covering such particular subject are not few.

In any case, we think that the places mentioned by Reviewer#2 should be reinforced with further cited literature, and the incorporation of references improves the review. We have added some new references in page 6. We did not add more because also the literature about the sustainable synthesis of MOFs under conditions compatible with the enzyme activity is still relatively scarce.

  1. In scheme 2, very specific enzyme and linker structures are used. Can authors specify the names? If they are referring the process to be more generic, then, more generic schemes/symbols should be used.

Although our intention was to prepare a scheme covering general strategy, Reviewer#2 is right. According to his/her advice, we have modified Scheme 2 to make it more generic.

  1. Table 2 is really well formed.

We also think so. Thanks very much.

  1. Figure 2 (d) does not have comparison with free enzyme. Can authors comment on this.

Whole Figure 2 (not only Figure 2d) was taken from literature (New J. Chem. 2018, 42, 4192–4200, doi:10.1039/c7nj03619a). None of us is co-author of such publication. It is true that, unlike Figs. 2a-c, where the activity of the immobilized enzyme in the MOF NH2-MIL-53(Al) is compared to that given for the free enzyme under different parameters (pH, temperature and reaction time), the catalytic activity of the enzyme supported on the MOF as a function of the store time at 4 ºC is not compared with the free enzyme. We suppose that the authors assumed that the free enzyme does not drop its activity within the 50 days tested for immobilized enzyme. We also would expect something like that, although we also think that such comparison should have been done.

In any case, we do not feel authorized to comment in our review anything beyond what the authors of that work commented, so we did not modify anything with respect the original version of the manuscript. Readers interested in those results could find that work adequately cited in our review, so they could contact the authors.  Despite everything mentioned above, we still think that Figure 2 gives interesting features about the activity and thermal, chemical and under-stored stability of the laccases one-pot-immobilized in NH2-MIL-53(Al).  

Reviewer 3 Report

The manuscript submitted to Catalysts entitled "Sustainable one-pot immobilization of enzymes on/in MOFs" by M. Asunción Molina and co-workers presents a concise short review carefully prepared in a subject with increased significance in recent years. The majority of relevant literature about this subject is referenced and properly discussed by the authors.

However, the conclusion would benefit if future perspectives and research directions could be included.

A few typos can be found throughout the text. For instance:

Line 113: “Talking” should read “Taking”

Line 143: A few abbreviations are not mentioned as their whole word. Cytc is one of these examples, but others can be found in the text.

Line 267: “enzyme.fre” should read “enzyme free”?

Thank you

Author Response

We are very grateful to Reviewer#3 for his/her carefully reading our manuscript and evaluating it in a constructive way.

Please find  detailed replies to any of his/her comment/suggestions in the attached word document. The changes in the new version of the manuscript have been introduced with the ‘Track Changes’ Tool of Word

Round 2

Reviewer 1 Report

  1. The manuscript has many grammatical mistakes. Line number 116- must give comma after ‘therefore’. This is just one example.  
  2. The author must send a manuscript to an English editing service and must submit the certificate while submitting your next response.
  3. I found two more spelling mistakes- Line 152- hydrophylic and 281- biocomoposites.

Author Response

All Referee1's comments in this second evaluation and most of those in the first evaluation are in the sense of prompting us to ask for an English editing service.

Obviously we are not native English speakers. However, over the course of our career we have published several dozen articles in English, and we have never received such insistence on using an English editing service.

We have discussed this point among all the authors and we have decided that we are not going to request this language editing service, due to the following reasons:

  1. After several decades writing articles in English, our command and control of this language (which has been always good enough for scientific publications) could not have worsened
  1. In the different evaluations that we have received about this manuscript, other Reviewers have highlighted that the manuscript is well written.
  1. Although the manuscript could have some grammatical errors seen from a purist point of view of English, we do not believe that it prevents the perfect scientific understanding of what we mean in every sentence.
  1. In the same way, it is true that the manuscript in its different versions may have had some spelling errors, the result of typing errors, not of ignorance of English language.
  1. All these errors must have been corrected or at least minimized in the different efforts we have made in revising the manuscript. As a consequence, the grammatical and spelling errors highlighted by Reviewer#1, as well as some others detected by us, have been modified.
  1. At the insistence of Reviewer#1, for the last revision, we have had the invaluable and selfless help of a native English speaker who has helped us to improve the English of the manuscript.
  1. These two grammar checks have been carried out during our respective vacation periods, and scrupulously complying with the response times given by the journal Catalysts (which are always short). Too much effort now to have to turn to a company that edits the English of a manuscript that, in our opinion, is already sufficiently improved.

Round 3

Reviewer 1 Report

The manuscript is well written with no grammatical mistakes.